# Estimating the Carbon Footprint of Healthcare in the Canton of Geneva and Reduction Scenarios for 2030 and 2040

**DOI:** 10.3390/ijerph21060690

**Published:** 2024-05-28

**Authors:** Bruno Mermillod, Raphaël Tornare, Bruno Jochum, Nicolas Ray, Antoine Flahault

**Affiliations:** 1Institute of Global Health, Faculty of Medicine, Université de Genève (UNIGE), 1202 Geneva, Switzerlandnicolas.ray@unige.ch (N.R.); 2Climate Action Accelerator, 1202 Geneva, Switzerland; 3Institute for Environmental Sciences, Université de Genève (UNIGE), 1205 Geneva, Switzerland

**Keywords:** carbon footprint, Geneva’s healthcare, carbon dioxide, greenhouse gas, healthcare system, sustainability

## Abstract

Switzerland, a wealthy country, has a cutting-edge healthcare system, yet per capita, it emits over one ton of CO_2_, ranking among the world’s most polluting healthcare systems. To estimate the carbon footprint of the healthcare system of Geneva’s canton, we collected raw data on the activities of its stakeholders. Our analysis shows that when excluding medicines and medical devices, hospitals are the main greenhouse gas emitter by far, accounting for 48% of the healthcare system’s emission, followed by nursing homes (20%), private practice (18%), medical analysis laboratories (7%), dispensing pharmacies (4%), the homecare institution (3%), and the ambulance services (<1%). The most prominent emission items globally are medicines and medical devices by far, accounting for 59%, followed by building operation (19%), transport (11%), and catering (4%), among others. To actively reduce Geneva’s healthcare carbon emissions, we propose direct and indirect measures, either with an immediate impact or implementing systemic changes concerning medicine prescription, building heating and cooling, low-carbon means of transport, less meaty diets, and health prevention. This study, the first of its kind in Switzerland, deciphers where most of the greenhouse gas emissions arise and proposes action levers to pave the way for ambitious emission reduction policies. We also invite health authorities to engage pharmaceutical and medical suppliers in addressing their own responsibilities, notably through the adaptation of procurement processes and requirements.

## 1. Introduction

In 2022, the Swiss healthcare sector played a central role in the country’s economic landscape, accounting for 11.3% of the country’s GDP. At the same time, it contributed significantly to the country’s environmental challenges, accounting for 6.7% of the country’s annual greenhouse gas (GHG) emissions, making it the third most polluting healthcare system per capita after the United States and Australia and ahead of Canada [1].

The current state of research in this area is informed by initiatives such as the “NHS Net zero initiative” [2], a UK national target to achieve zero GHG emissions in healthcare by 2040, and “The Shift Project” [3], which has assessed the carbon footprint of the French healthcare sector. These global endeavours underline the importance of considering the environmental impact of healthcare. With the NHS Net Zero Initiative, the British National Health Service (NHS) has committed to becoming carbon neutral by 2040. This means that the NHS aims to reduce its GHG emissions to net zero by this date by implementing measures such as improving energy efficiency, utilising renewable energy sources, and reducing waste. The NHS is responsible for around 5.4% of the UK’s total GHG emissions. By comparison, in 2015, the NHS produced around 25.8 million tonnes of carbon dioxide equivalent (MtCO_2_e), of which 20% was from medicines and chemicals, 10% from building energy, 10% from medical equipment, 5% from anaesthetic gases and inhalers, 5% from waste, and 24% from the other supply chains. This includes emissions from various sources, such as energy consumption in hospitals and clinics, the transport of patients and staff, the procurement of medical goods and medicines, waste management, etc.

The Shift Project is a French think tank focussing on the transition to a low-carbon economy. It has studied the environmental impact of the French healthcare sector and found that this sector accounts for over 8% of the country’s carbon footprint and emits around 49 million tonnes of CO_2_ equivalent annually. The main contributors include medicines (29%), medical devices (21%), food (11%), transport (9%), and heating (9%). To change this, The Shift Project is committed to energy efficiency, renewable energy, and sustainable procurement in the healthcare sector.

Despite having one of the most expensive healthcare systems in the world and one of the highest life expectancies in Europe and the world, Switzerland faces a multi-faceted challenge. Its healthcare system must maintain the quality of care while improving preventative healthcare and reducing its carbon footprint without driving up healthcare costs. Healthy life expectancy at the age of 65 in Switzerland is several years behind that of Sweden or Japan, indicating gaps in preventive measures. Studies show that effective prevention enables people to live longer without serious disabilities [4,5].

The efficiency of the Swiss healthcare system, measured by the quality/energy footprint indicator, is below optimal standards, especially when compared to countries such as Sweden [6]. Sweden achieves a comparable quality of healthcare with an energy footprint of one-third, indicating potential for improvement. Interestingly, the healthcare sector in Switzerland, including in the canton of Geneva, has not committed to actively reducing its carbon footprint until recently. Traditionally, the focus has been on improving the quality of care, patient safety, and cost containment, while little attention has been paid to environmental sustainability.

An important finding from a previous study published in 2019 by Belkhir et al. [7] emphasises the environmental impact of medicine products. Despite increased global efforts to curb carbon emissions, little attention has been paid to the healthcare system, particularly the pharmaceutical industry. Belkhir et al. [7]. concluded that this industry is much more emissions-intensive than the automotive industry, challenging conventional assumptions about its environmental impact.

Our initiative in the Canton of Geneva (a term used to describe a member state of the Swiss Confederation), which is driven by the need to maintain and improve the performance of the healthcare sector, particularly in terms of efficiency, aims to address three main challenges. Firstly, we endeavour to gather knowledge and identify the main sources of carbon emissions in the healthcare sector. Secondly, we estimate the carbon impact of the healthcare system in the Canton of Geneva. Finally, we propose strategic guidelines to reduce this carbon footprint based on future projections following IPCC projections for 2030 and 2040 and taking into account the expected challenges.

## 2. Materials and Methods

Our approach to assessing the carbon footprint of the Canton of Geneva’s healthcare system is based on collecting data from the key players in the system. We sought collaboration with hospitals and clinics, both public (1 out of 1) and private (3 out of 7, 67% of beds), nursing homes (480 beds out of 4000), pharmacies (32 out of 181), home care (1 out of 1), and ambulance services (10% of ambulances) (see Figure 1).

We interviewed each stakeholder relevant to the study to collect their data (see Figure 2) so that we could assess the specific carbon footprint of their activities according to the Greenhouse Gas Protocol [8]. Unfortunately, the results of analysis laboratories could not be integrated directly, due to the lack of cooperation from this sub-sector. However, an estimate was made using economic data and monetary factors for medical activities. For medical practises, we relied on data from Pr. Nicolas Senn, who recently conducted a study on the ecological design of medical practises in French-speaking Switzerland, due to their heterogeneity and the very different data quality. We had also planned to include the Department of Health of the State of Geneva in our study, but we were unable to obtain data from it. Its offices are shared with other government departments.

When specific data were not available, we used similar data from the same type of stakeholders. For example, to determine the carbon footprint of anaesthetic gases from a private hospital that had not provided data on this topic, we extrapolated it based on the hospital activities of another hospital. This hybrid method, combining bottom-up and top-down data collection, allows us to obtain a complete picture for the estimation of the carbon footprint of the Geneva hospital system, while remaining as close as possible to the specific characteristics of each actor. The Figure 3 illustrates the system boundary diagram, providing an overview of the various sources of CO_2_ emissions associated with a healthcare institution. The diagram categorises emissions into three scopes: Scope 1 includes direct emissions from sources controlled by the institution, such as heating, fleet of vehicles, medical gases, bronchodilators, and refrigerant gas leaks. Scope 2 encompasses indirect emissions from purchased electricity and district heating. Scope 3 accounts for other indirect emissions, including those from incoming materials, upstream and downstream freight, and business travel. This comprehensive visualisation is crucial for understanding the institution’s carbon footprint and identifying areas for effective emission reduction strategies.

The raw data collected were then converted into equivalent tonnes of CO_2_ (tCO_2_eq) (Figure 4). In most cases, the data were presented in the form of quantities, such as kilometres driven, CHF, kWh, and MJ. Our methodology is inspired by two studies that also aimed to estimate the carbon footprint of a healthcare system at a broader range, the NHS Net zero initiative for the British healthcare system and The Shift Project for the French one.

We used pre-calculated emission factors (EFs) (see Table 1), although these are subject to some uncertainty, as they are often based on national or even global averages. However, this method is the most commonly used, as it is often not possible to systematically perform a full life-cycle analysis for each emission point. We used the EcoInvent 3.9 database, which we queried using the OpenLCA 2.0 software [10,11] (Table 1). Depending on the activity, this database provides global data or data specific to Switzerland. We also used the Mobitool 3.0 database [12] (in which the EcoInvent emission factors were largely incorporated), the standard for the environmental assessment of means of transport and mobility in Switzerland.

Unfortunately, there are very few specific EFs for each medicine. For some that have been identified as highly harmful to the environment, full life-cycle studies have been conducted and are available in the literature. This is the case for certain anaesthetic gases and bronchodilators, allowing us to quantify their effects more precisely [13,14].

For the other medicines, we used a monetary EF calculated by the French Environment and Energy Management Agency based on EEIOT (Environmentally Extended Input Output Table) economic modelling [15]. A monetary EF is an indicator that measures the amount of GHG emissions produced for each monetary unit generated by an economic activity. It helps to assess the environmental efficiency of activities by relating the economic value to carbon emissions. As we did not find any data in this area in Switzerland, we relied on the existing French database “Base Empreinte” [15]. We assumed that the values determined were sufficiently close to those of the Canton of Geneva, as this sector is highly globalised, and around 80% of active ingredients are manufactured in India and China [16].

For the fuels category, the KBOB tool of the Swiss Confederation (IPB 2009/1:2022, version 2) provided us with specific EFs for Switzerland [17]. Finally, electricity in Geneva is already highly decarbonised, with more than 30% of electricity coming from local renewable sources and around 70% from the Swiss electricity mix, which is also low carbon compared to other countries, such as Germany. The EF was calculated jointly by the Geneva institutions, OCSTAT, OCEN, SIG, and Direction Durabilité et Climat (DDC).

**Table 1 ijerph-21-00690-t001:** Databases used for the carbon footprint calculation.

Source of Emission Factors	Category of Raw Data
EcoInvent 3.9 [10]	Food, purchases, wastewater treatment, waste
Mobitool 3.0 [12]	Transports
BaseEmpreinte V23.1 [15]	Monetary EF for medicines and medical activities
The Shift Project Excel Chiffrage 2023 v1.0 [18]	Monetary EF for medical devices
KBOB/IPB 2009/1:2022, Version 2 [17]	Fuels and combustibles
OCSTAT, OCEN, DDC, SIG 2022 [19]	Specific electricity EF for Geneva
Andersen et al., 2012 [20]	Anaesthetic gases (sevoflurane, desflurane, isoflurane)
Janson et al., 2020 [14]	Bronchodilators
Parvatker et al., 2019 [13]	Anaesthetic gases

Next, we modelled emission reduction scenarios based on the joint socio-economic projections published by the IPCC [21] for the year 2023 (Table 2). These scenarios, based on the carbon footprint of the Geneva health system for 2022, are quantified targets categorised to be consistent with the IPCC trajectories. They do not take into account the pressures and increasing demand in the health sector due to the ageing population and the growing number of people with lifestyle-related diseases. The first scenario does not require any special measures from the healthcare system, and we call it “business as usual”. It is based on the IPCC’s SSP2-4.5 scenario. Under this scenario, emissions remain stable in 2030 and 2040 despite population growth and ageing, as the carbon footprint of the healthcare system is mitigated by the gradual decarbonisation of the entire economy and society, from electricity, heating, and infrastructure to mobility and dietary behaviour. The second scenario shows the impact of “modest” measures on GHG emissions. It is based on the SSP1-2.6 scenario. The third scenario is based on coordinated and “ambitious” measures by all stakeholders in the canton and is modelled on the IPCC’s SSP1-1.9 scenario.

Finally, we proposed measures to reduce the carbon footprint of Geneva’s healthcare system. These measures emerged from the literature, from our interviews with stakeholders during our data collection, and from examples such as The Shift Project and the NHS that have proposed similar measures. Therefore, their reduction potential is qualitative rather than quantitative, as these are our estimates. Ease of application and impact are rated on a star scale. The fewer stars there are, the more difficult it is to implement these measures and the less impact they will have on reducing the carbon footprint. The reduction potential category represents the targets we propose for each category in the ambitious scenario by 2040. These targets are not quantified on the basis of individual measures but as targets to be achieved for each main category.

## 3. Results

This study presents trends and estimates rather than precise measurements. Uncertainties arise mainly from the emission factors, in particular, the monetary emission factor applied to items such as medicines and medical devices. Further details on these limitations can be found in the “Discussion” section. We were forced to enter into non-disclosure agreements (NDAs) with numerous stakeholders to ensure the confidentiality of their data. Consequently, the data presented in this results section are the summary of their data in percentages. Our overall estimate of the carbon footprint of the Geneva healthcare system within the given framework amounts to 436,831 tonnes of CO_2_ equivalent for the year 2022, which corresponds to approximately 0.28% of Switzerland’s total territorial and consumption-related carbon footprint for 2021 (118.68 + 35.79 million tonnes CO_2_eq). As there is no carbon footprint for the canton of Geneva, we were unfortunately unable to make this comparison at the cantonal level. Our results make it possible to categorise greenhouse gas emissions by the healthcare sector. In an initial analysis, we provisionally excluded medicines and medical devices, in line with The Shift Project’s presentation of the breakdown of emissions by actor. We ended up with the hospitals sub-sector in first place, accounting for 47% of the sector’s emissions (Figure 5). This is followed by nursing homes and medical offices, with 20% and 18%, respectively.

Finally, the analysis laboratories, pharmacies, home care institution, and ambulance services each accounted for less than 10%. These results are consistent with those of The Shift Project. The Shift Project estimates that, in France, hospitals account for 46% of the carbon footprint, outpatient medicine for 28%, and home care institution for 25%, excluding facilities for the disabled and health insurance companies [3]. If we use the same grouping as The Shift Project in the Canton of Geneva—i.e., grouping laboratories, pharmacies, and medical offices together in a category called “outpatient medicine”—we obtain 29% of the carbon footprint, and if we group nursing homes and home care institution together, we get 23%, which is similar to The Shift Project values with a similar scope. The detailed breakdown by CO_2_-emitting activities for each actor of the healthcare system can be found in Appendix A (Appendix A).

In a second analysis, where we include medicines and medical devices, the situation is quite different. In our study, the term “pharmacies” refers exclusively to retail pharmacies, with the exception of hospital pharmacies, whose business model is mainly based on the sale of medicines. While the carbon footprint of medicines can only be indirectly attributed to pharmacies, they are at the top of the list of GHG emissions for the purchase of medicines, ahead of hospitals (Table 3 and Figure 6).

Finally, we present the distribution of GHG emissions within the Geneva healthcare system in descending order to illustrate the predominant impact of the various activities on the overall carbon footprint. At the top of the list are medicines and medical devices, which account for 58.6% of the total carbon footprint. Buildings and their operation follow with 18.8%, emphasising the importance of sustainable practises in the design and operation of medical infrastructures. Commuting and professional transport account for 10.8% of emissions, while food (4.1%), laundry (2.3%), and the purchase of medical goods (1.8%) also make a significant contribution. Waste, which includes various forms of waste generated by healthcare activities, accounts for 1.5%. Non-medical purchases, anaesthetic gases and bronchodilators, and IT complete the list at 0.9%, 0.8%, and 0.4%, respectively (Figure 7).

A table summarising the measures was developed. These measures were proposed after our interviews with the stakeholders of our data collection, and their reduction potential is only a qualitative estimate. Table 4 lists the measures, the ease with which a measure can be applied, its potential to reduce the carbon footprint, and the organisation(s) responsible for its application.

The modest effort scenario by 2030 (Figure 8) is in line with the IPCC SSP1-2.6 scenario and aims to reduce GHG emissions from the healthcare sector by 21%. Direct measures alone would help to reduce the carbon footprint by 7%, mainly thanks to buildings and infrastructure. Indirect measures and measures in the area of medicines would make it possible to reduce the gap to the 21% target.

We decided to combine the scenario with the ambitious efforts for 2030 and the modest efforts for 2040, as their targets are close to each other, namely −43% for 2030 and −46% for 2040 (Figure 9). We therefore chose −45% as the target and created this scenario with the various reductions broken down by item. In this scenario, direct measures would contribute to a 15% reduction in the carbon footprint, while medicines and indirect measures would make it possible to achieve the final target of −45%.

The last scenario (Figure 10) is probably the most convincing because its timeframe allows real long-term action plans and the impact of certain indirect measures to begin to take effect. We rely on the SSP1-1.9 projection with the long-term goal of achieving carbon neutrality by 2050 in order to limit global warming to +1.5 °C.

Again, it will be possible to reduce the carbon footprint by −23% through direct action alone, mainly from heating buildings and transport. We assume that, by 2040, medicines will only be produced and transported using low-carbon energy sources and that their overconsumption will be fully under control. We also assume that our proposals for indirect measures such as prevention and health promotion, the digitalisation of the healthcare system, and the introduction of active transport and healthy eating within planetary boundaries will help to reduce the carbon footprint by 69% or more.

## 4. Discussion

It is interesting to compare our results with the various projects carried out in other health systems (Table 5). Although the Geneva system is the smallest, it is a state-of-the-art healthcare system that is certainly comparable to others around the world. Compared to France, whose Shift Project initiative is closest in methodology and scope to ours, we see that Geneva’s carbon footprint per capita is more than 16% higher, from 0.73 to 0.84 tCO_2_eq. These figures should be set in relation to the study by Andrieu et al. published in 2023, which shows that the energy footprint of the Swiss healthcare system is more than three times larger than that of France [6]. In the UK, the per capita footprint is significantly lower than in Geneva. This result can be explained by a fully public healthcare system, a different study framework and the fact that discussions about reducing the carbon footprint only started in 2008. The other studies focusing on the healthcare systems in Quebec, Portugal and Australia were conducted with a pure top-down approach without collecting raw data from the healthcare facilities, which is different from the method we chose in this article [22,23,24].

Taking our findings into account, direct and indirect measures to curb GHG emissions can be identified. For each of the emission items, we develop specific levers for action. These direct levers include areas such as optimising the energy efficiency of buildings and means of transport that favour low-GHG mobility. They also include changes in culinary habits—with the introduction of a diet served to nursing homes’ staff and residents that respects planetary boundaries—and waste management based on the 3Rs (Reduce, Reuse, and Recycle) principles. These direct actions also include the substitution of commonly used anaesthetic gases, such as desflurane and nitrous oxide, which are major sources of greenhouse gases, and the replacement of bronchodilator aerosols with powder formulations whenever possible (and acceptable to the patient). Finally, our direct actions include proposing (inter)national incentives to make medicines less carbon-intensive and to encourage the recycling of part of their production. In particular, sending economic signals and adapting procurement criteria and processes by relevant stakeholders to favour low-carbon suppliers and alternative products could contribute significantly to the necessary decarbonisation of the healthcare system.

At the same time, we identified indirect levers that act as catalysts for other, more direct measures. The coordination of hospitals to consolidate infrastructure and collective cost management, combined with specific training initiatives to promote awareness of sustainable healthcare practises among professionals, would likely contribute to a significant reduction in carbon emissions. In addition, incentivising research in this area; streamlining digital processes to enable personalised precision medicine; reducing healthcare waste; and promoting preventative measures in the areas of diet, tobacco, alcohol, and physical activity can further contribute to this goal. Each of these levers would help to achieve the goals outlined in our scenarios and drive the healthcare system towards a more sustainable and resilient way to minimise its carbon footprint.

The measures proposed in this study are based on best practises for reducing the carbon footprint of the healthcare system, derived from a thorough literature review and discussions with stakeholders. Their applicability depends on the different characteristics, needs and actions of the various stakeholders. Many of the proposed measures offer added value by having a positive impact on both public health and the environment, while contributing to the economic sustainability of those implementing the measures. Each recommended measure is closely modelled on the guidelines of the NHS Net Zero initiative in the UK and The Shift project in France.

Several direct measures are proposed for buildings, including the implementation of efficient thermal refurbishments that meet the highest standards. Switching heating and cooling systems from fossil fuels to sources with a lower carbon footprint would significantly reduce GHG emissions associated with heat production, as would the gradual replacement of refrigerant gases with environmentally friendly air conditioning systems.

In the transport sector, the focus is on increasing the share of active modes of transport, such as cycling and walking, which are beneficial for both health and the carbon footprint. A Swedish study states that an investment of EUR 100 million between 2018 and 2030 in cycling infrastructure in Stockholm would lead to an annual saving of EUR 12.5 million in health costs, simply by increasing physical activity [25]. It is also recommended to promote the use of public transport and car sharing. In addition, the partial teleworking of administrative staff could be encouraged, if this is not already the case. The vehicle fleet of companies could be gradually converted to electric motors. It is also recommended to limit air travel by executives to conferences while promoting e-learning and video conferencing and encouraging the use of rail wherever possible. In addition, the development of telemedicine would help to limit non-essential patient travel. Finally, reducing the number of vehicles emitting greenhouse gases and particulate matter (of all kinds) would not only help to reduce the carbon footprint but also improve respiratory health by reducing exposure.

In terms of catering and food consumption in everyday life, the aim is to follow the “healthy planet” dietary recommendations of the EAT-Lancet Commission [26]. This diet, which is not only beneficial for the environment, should prevent the death of 10.9 to 11.6 million people per year worldwide. Specifically, a healthier diet means reducing red meat and added sugars by more than 50% (compared to the current Western diet) and doubling the portions of vegetables, fruit, and nuts. In addition, tackling food waste in the catering industry, estimated by The Shift Project to be around 20%, would have a significant impact on carbon emissions.

Waste must be reduced at source. One of the most important measures is therefore to follow the 3R rule (Reduce, Reuse, and Recycle) to avoid the production of incinerable waste. A pilot study conducted at Lausanne University Hospital in Switzerland on anaesthesia waste has shown that medical waste, whose incineration has three times the impact of household waste [15], can be reduced by 85% after proper sorting [27]. As far as medical devices are concerned, reuse should be promoted instead of the use of single-use devices by encouraging the manufacture and use of reusable medical devices under conditions that ensure patient safety and quality of care.

Numerous studies have shown that anaesthetic gases have a significant impact on the environment. It has been suggested that the use of anaesthetic gases with a high greenhouse effect, such as desflurane and nitrous oxide, should be banned and replaced by sevoflurane or other alternatives [20]. Desflurane and nitrous oxide have a 100-year global warming potential that is 2540 and 273 times that of CO_2_, respectively, compared to 130 times that of sevoflurane. Nitrous oxide is not only used in anaesthesia but also frequently in outpatient treatment. It would be possible to reduce GHG emissions from anaesthetic gases to virtually zero if desflurane and nitrous oxide were replaced by alternatives [28]. In addition, several studies recommend performing more intravenous anaesthesia instead of using gas [29]. Finally, it is recommended to systematise the use of dry powder-propelled bronchodilator inhalers whenever possible, or propellant gases, which have a low environmental impact and emit up to 28 times less GHG than conventional gas inhalers [30].

To minimise the carbon footprint of medicines and medical devices, it is suggested that manufacturers are encouraged to reduce the unit carbon cost of each medicine by taking action in regard to the manufacturing processes and the carbon intensity of the energy used. It is also recommended to reduce the waste of medicines and medical devices. Health institutions can also play a role in sending signals to manufacturers by developing procurement criteria and adapting tendering procedures (e.g., by requiring transparency on the carbon value of items, disclosure of the supplier’s carbon footprint, adoption of ambitious decarbonisation plans, etc.). We could also hope to reduce the amount of wasted medication through better coordination between doctors and nurses using IT tools and the introduction of electronic patient records, including access to the pharmacist. The availability of medication prescriptions by unit in pharmacies would also help. Solutions should be sought to sensitise not only patients but also the medical and nursing professions to the need to manage healthcare responsibly and to promote more targeted prevention, diagnosis, and treatment practises in order to avoid unnecessary overuse. These nationwide incentives, coupled with concrete commitments, would create significant pressure for greener practises, while ensuring the continued availability of medicines that are essential to the health of the population.

Indirect levers are cross-cutting measures that would act as catalysts and enable the activation of future levers. Relying solely on the direct levers mentioned above will probably not be enough to achieve the ambitious targets for reducing GHG emissions, and measures are needed that truly address healthcare processes and professions. Through these various measures, it would be advisable to reflect on the role of the healthcare system, which is more focused on keeping the population in good health and less on waiting for diseases to develop that then need to be treated. This virtuous circle requires greater promotion of preventive healthcare in order to avoid costly medical treatments with high follow-up costs. The use of emergency services and hospital treatment could be minimised as much as possible, as they make a very large contribution to the carbon footprint of the cantonal healthcare system. A far-reaching and gradual reorganisation of the healthcare system, particularly with regard to access to care, could be implemented. In Denmark, for example, patients have to go to a general practitioner, who then refers them to a specialist or for further investigations or even to a hospital. This reform has made it possible to reduce the number of hospitals from 128 to 21 in forty years [31]. These changes, which are beneficial for the sustainability and resilience of our healthcare system, would need to be managed at the state level. This would require the support and training of healthcare professionals, more expertise in these issues, the use of digital technology to support process efficiency, and the promotion of preventive behaviours conducive to better population health.

If we focus on prevention and health promotion upstream, we can improve the health of Geneva’s population and reduce their use of healthcare services. The challenge is to better control the demand for treatment and travel; reduce the use of infrastructure; consume less medication; and use less medical equipment, imaging, and biological analyses, all in the service of more effective and efficient medicine. An analysis of healthcare expenditure shows that Switzerland spends less than 3% of its healthcare expenditure on prevention and health promotion, which is below the average for OECD countries [32]. The desire to reduce the environmental footprint of the healthcare system could prove to be an opportunity for stakeholders to consider prevention as a real investment. The return on investment (ROI) for each franc (CHF) invested in tobacco control is between CHF 28 and CHF 48, and between CHF 11 and CHF 29 for alcohol control [33].

Switzerland faces significant costs related to addiction to tobacco, alcohol, and illicit medicines, amounting to CHF 3 billion, CHF 477 million, and CHF 274 million, respectively, in 2017 alone [34]. The direct costs to the healthcare system attributable to patients suffering from these addictions are considerable and amount to almost CHF 4 billion. It should be noted that our EF estimates that for every CHF 1000 spent on services and activities related to human health, approximately 100 kg of CO_2_ is emitted [15]. Based on the Geneva population and assuming that the Geneva population is not more affected by these addictions than the rest of the Swiss population, the direct cost of these addictions to the canton’s healthcare system would amount to nearly CHF 221 million per year. Multiplied by the emission factor used in our study, this would correspond to emissions of 22,000 tCO_2_eq from these addictions alone. Prevention aimed at reducing these risky behaviours, both for the individual and for the planet, would make it possible to reduce the use of the healthcare system and the carbon footprint of the associated care.

Promoting active mobility, such as cycling and walking, is good for health and the environment [25]. Reducing the number of vehicles that emit greenhouse gases and particulate matter and increasing the share of active mobility not only help to reduce the carbon footprint but also improve people’s health by reducing their exposure to pollution and increasing their physical activity. Promoting moderate daily physical activity, healthy eating (Planetary Nutrition [26]), intellectual practises, and socialisation limits many chronic diseases (e.g., diabetes, obesity, cardiovascular disease, cancer, and Alzheimer’s disease), and reducing alcohol and tobacco consumption would lead to large health gains and substantial savings, while also reducing the carbon footprint of the healthcare system. Our analyses show how important it is to invest in prevention, especially in terms of public health, but also in terms of economic efficiency and environmental sustainability.

Our study enabled us to decipher the main sources of carbon dioxide emissions in the Geneva healthcare system. In the medical context, medicines and medical devices play a prominent role and even overshadow other emission items in hospitals. Unsurprisingly, we also find the same main emission items as in other sectors, namely the heating of buildings, commuting, and food. When comparing the different sub-sectors of healthcare, hospitals are at the top and are responsible for almost 50% of the healthcare sector’s total carbon footprint. On the one hand, this shows the responsibility of hospitals and public health policy to reduce their impact, which has already been emphasised in previous studies. Contrarywise, it raises awareness of the other, more heterogeneous actors who are responsible for the other half of emissions and who should share the goal of reducing emissions overall.

Our study on the carbon footprint of the healthcare hospital system was intended to be as representative as possible of the GHG emissions emitted by this system in the canton of Geneva and to provide trends and estimates rather than precise measurements. However, it is important to recognise several limitations that may have affected the accuracy and sometimes the reliability of some of our results.

Firstly, the lack of direct GHG measurements is a limitation that this type of study has to contend with. The collection of raw data, which were subsequently converted to tCO_2_eq, using emission factors, and the occasional use of monetary units (CHF) to quantify GHG emissions, particularly for medicines and medical devices, may have introduced uncertainties. The monetary factor used for medicines is independent of the type of medicine, which does not necessarily reflect the reality of emissions associated with each medicine, but it hopefully gives a representative average. We used the French emission factors for medicines and medical devices, although prices in Switzerland are different and may introduce uncertainties. To convert the EFs for medicines and medical devices into Swiss francs, we multiplied the EFs by the conversion rate of 2018, the date of calculation of these EFs, and then accounted for inflation until 2022. In addition, the monetary EFs are inherently subject to a very high degree of uncertainty [15], estimated by The Shift Project to be 80% for medicines and 50% for medical devices. The results of calculations using these factors should therefore be treated with caution. Emission factors based on raw data are much more accurate, with uncertainties ranging from a few per cent for heating, for example, to around 60% for certain modes of transport per km [15].

Secondly, it should be noted that neither the NHS Net Zero Initiative nor The Shift Project made final calculations of the uncertainty interval for their carbon footprint results until 2022. In 2023, The Shift Project arrived at an uncertainty interval between 6.6 and 10% of its carbon footprint. This project uses a lot of estimates, averages, and extrapolations at a national level. We expect to achieve a similar level of uncertainty intervals by using few monetary EFs and few subjective estimates and collecting a large amount of data on the ground. Finally, this study is limited to GHG emissions, and further research that includes a life-cycle environmental impact assessment should be integrated to refine our results. Although it is important to be aware of these limitations when interpreting the results of our analyses, the conclusions from this study are in line with those of the international literature on the subject. More than the precision of the figures, the main value lies in showing the extent of emissions, the magnitudes, the relative contribution of each source, and the identification of levers and solutions.

## 5. Conclusions

In conclusion, this study addressing GHG emissions, the first of its kind in Switzerland, observes a similar distribution of GHG emission in the healthcare system as those in France or the United Kingdom These primary results highlight the prominent role in GHG emissions played by medicines and medical devices that are at the heart of the healthcare services. Other very significant emissions come from categories not directly related to medical practise, namely heating, transport, catering, and laundry. Our different scenarios hope to pave the way for key stakeholders to design and adopt ambitious health policies that include environmental considerations, improvements in health prevention, and quantified time-bound decarbonisation pathways. With this work, we are calling for the establishment of an institutionalised system for monitoring Geneva’s public health emissions that can progressively address methodological limitations as part of a continuous improvement process.

## Figures and Tables

**Figure 1 ijerph-21-00690-f001:**
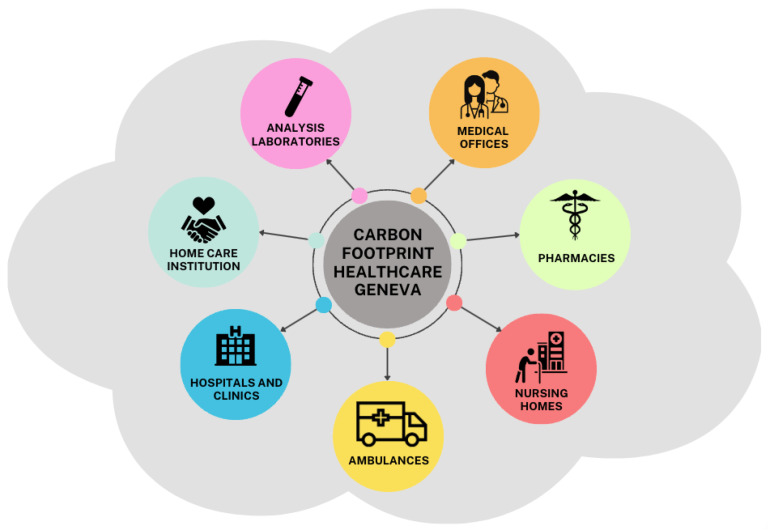
The different players in the healthcare system included in our study.

**Figure 2 ijerph-21-00690-f002:**
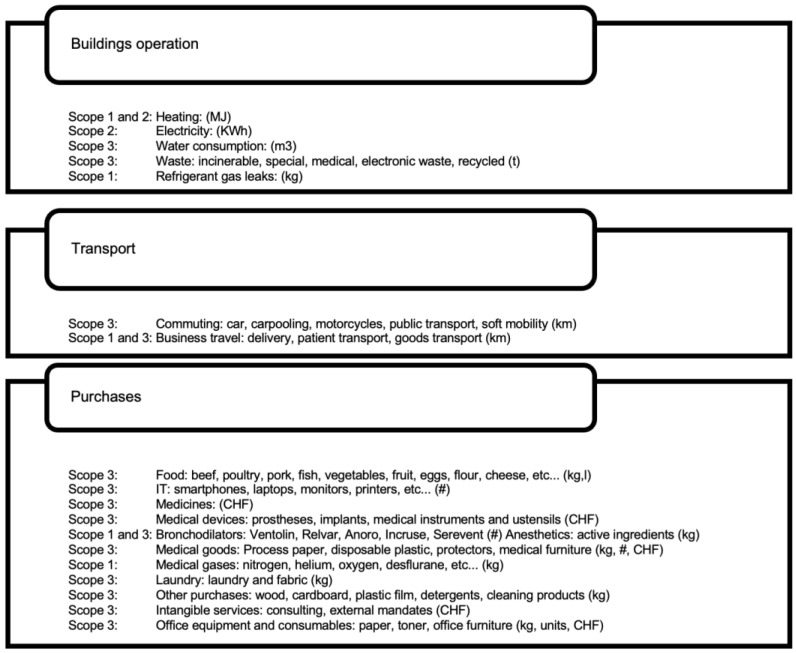
Data collected from each institution and their units. Scopes 1, 2, and 3, as defined by the GHG protocol.

**Figure 3 ijerph-21-00690-f003:**
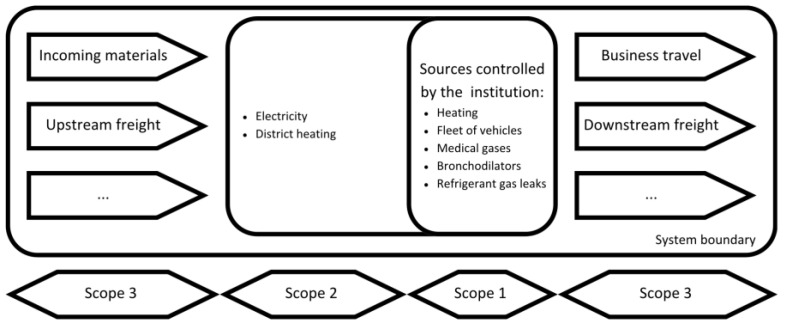
System boundary diagram summarising the different sources of CO_2_ emissions attributed to a healthcare institution (adapted from the French methodology for carrying out greenhouse gas emission assessments [9]).

**Figure 4 ijerph-21-00690-f004:**
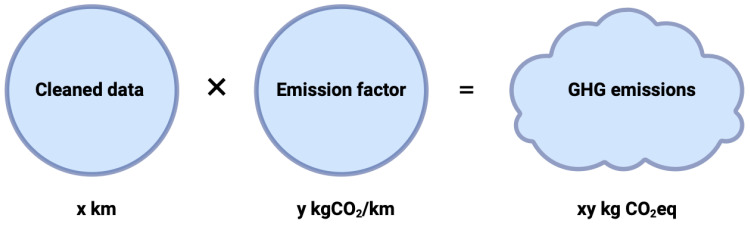
Method for calculating the carbon footprint of an activity. Here is presented an example with kilometres travelled.

**Figure 5 ijerph-21-00690-f005:**
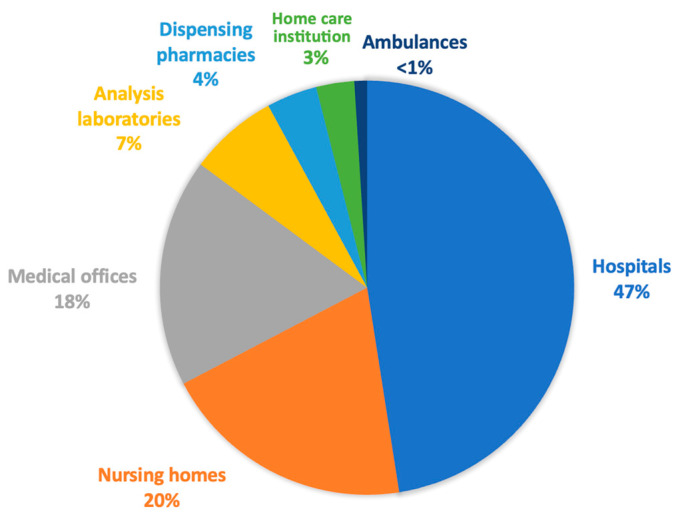
Distribution of the greenhouse gas emissions, excluding medicines and medical devices.

**Figure 6 ijerph-21-00690-f006:**
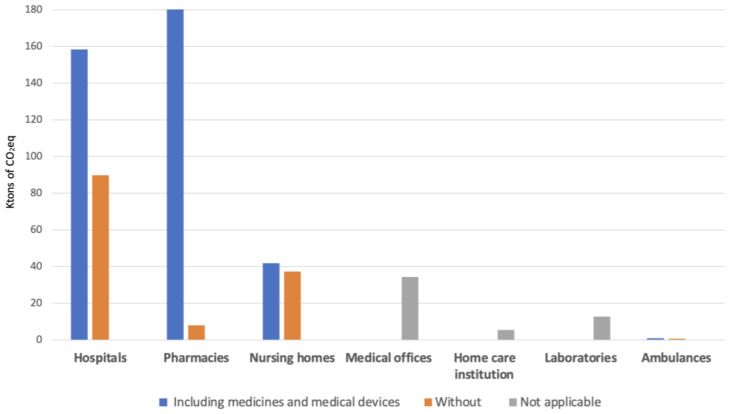
Distribution of the GHG emissions in kilotons of CO_2_eq.

**Figure 7 ijerph-21-00690-f007:**
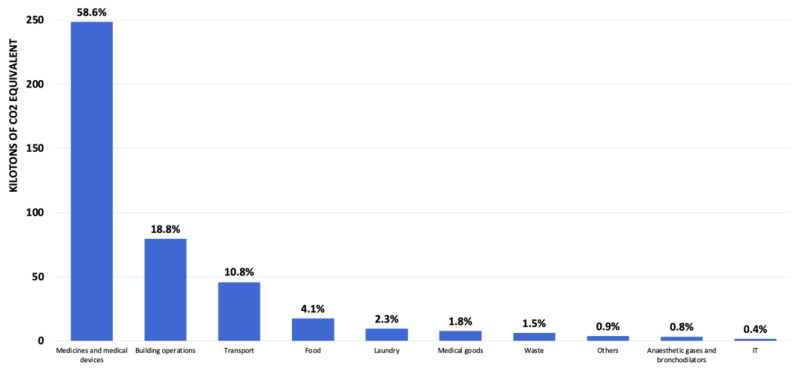
Distribution of the GHG emissions per activity.

**Figure 8 ijerph-21-00690-f008:**
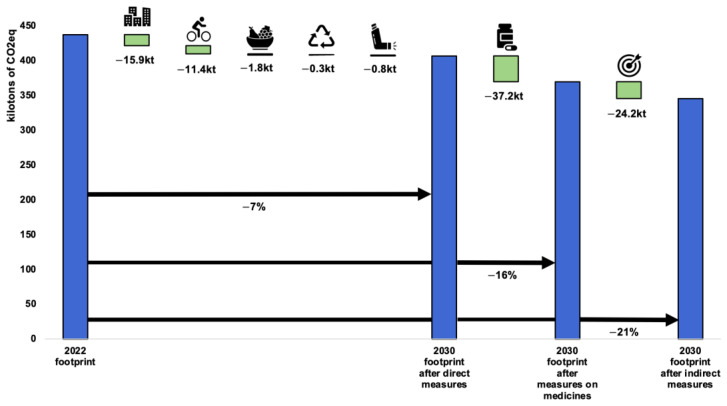
Modest 2030 scenario, quantified by emissions category. Categories: buildings, transport, food, waste, anaesthetic gases and bronchodilators, medicines and medical devices, and indirect levers.

**Figure 9 ijerph-21-00690-f009:**
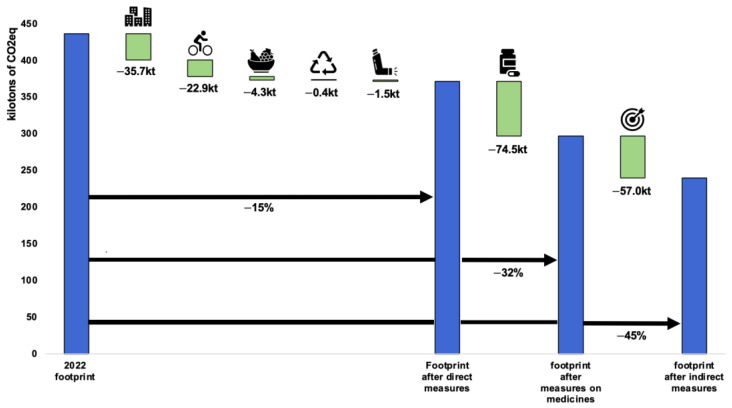
Ambitious by 2030 or modest by 2040 scenarios, quantified by emission category. Categories: buildings, transport, food, waste, anaesthetic gases and bronchodilators, medicines and medical devices, and indirect levers.

**Figure 10 ijerph-21-00690-f010:**
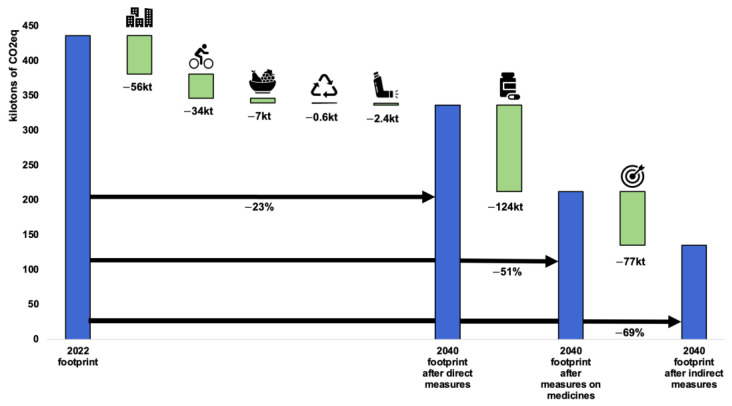
Ambitious 2040 scenario, quantified by emissions category. Categories: buildings, transport, food, waste, anaesthetic gases and bronchodilators, medicines and medical devices, and indirect levers.

**Table 2 ijerph-21-00690-t002:** GHG emission reduction scenarios, according to the IPCC.

Name	IPCC Scenario	Reduction by 2030	Reduction by 2040	Warming Limit
Business as usual	SSP2-4.5	-	-	<3 °C
Modest	SSP1-2.6	−21%	−46%	<2 °C
Ambitious	SSP1-1.9	−43%	−69%	<1.5 °C

**Table 3 ijerph-21-00690-t003:** GHG emissions calculated for each sub-sector in tons of CO_2_eq, with and without medicines.

Name	With Medicines	Without Medicines	Not Applicable
Hospitals	158,546	89,843	-
Pharmacies	183,227	7829	-
Nursing homes	41,592	37,310	-
Medical offices	-	-	34,314
Homecare institution	-	-	5336
Analysis laboratories	-	-	12,760
Ambulances	807	734	-

**Table 4 ijerph-21-00690-t004:** List of proposed measures to reduce the carbon footprint of the Geneva healthcare system, rated by ease of implementation and impact on emissions reduction (1 to 3 stars), based on discussions with relevant stakeholders, our estimates, and the projections of The Shift Project and the NHS. For each measure, the organisation responsible for implementation is also indicated. The ease of implementation ranges from minimal resource requirements to significant changes, while the impact ranges from marginal to significant emissions reductions. Brief descriptions accompany each measure for context.

Measures	Ease of Application	Impact	Reduction Potential	Who Is Responsible for Implementation?	Brief Description
**Buildings**	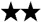	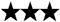	−70%	Policies and owners	Renovation to increase the energy efficiency of buildings and replace heating with low-carbon alternatives
**Transport**	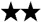	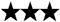	−75%	Institutions, policies and population	Encouraging employees and vehicle fleet managers to use less carbon-intensive and/or active forms of mobility
**Diet**	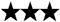	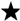	−40%	Institutions	Less meat on cafeteria and patient menus
**Waste**	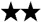	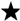	−10%	Institutions	Reduce waste by working with the committees in charge of medical protocols and producers of medical equipment
**Anaesthetic and medical gases**	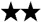	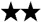	−60%	Hospitals, politicians, the pharmaceuticalindustry and doctors	Replace high-impact anaesthetic gases as far as possible (e.g., desflurane and N_2_O). Make marketing authorisation conditional on publication of the medicine’s carbon emissions.Partial relocation of the production of certain essential molecules in EuropeSobriety in prescribingSingle unit distributionReducing waste
**Bronchodilators**	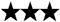	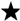	−70%	Doctors	Replace high-impact propellants with powder propulsion wherever possible
**Medicines**	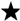	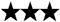	−70%	Politicians, the pharmaceutical industry and doctors	Reduce waste, including better coordination with IT medical records, introduce prescription by unit, raise professionals’ & patients’ awareness of medicines impact, incentivise the pharmaceutical industry, requirement on producers to comply with ambitious decarbonisation targets
**Grouping hospitals**	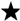	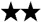	N/A	Policies	Pooling infrastructure, equipment and knowledge
**Research and formation**	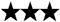	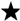	N/A	Institutions	Encouraging research into the ecological transition in healthcare Raising awareness and training staff in sustainability
**Digitalisation**	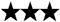	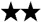	N/A	Policies	Implementing an information system with interoperability at all levels of the healthcare system, controlling data volumes(e.g., electronic patient record)
**Prevention and health promotion**	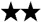	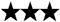	N/A	Institutions and policies	Move from a healthcare system based on “cure” to one based on “care”Promote preventive healthcare to a greater extent, so as to avoid costly medical treatment that has a high impact on subsequent emissions

**Table 5 ijerph-21-00690-t005:** International comparison of the carbon footprint of the healthcare system in tons of CO_2_ equivalent per inhabitant.

Region Concerned—Project	Year of Observation	MtCO_2_eq	tCO_2_eq/Capita	Perimeter
**France—The Shift Project**	2023	49.1	0.73	Public and Private
**Geneva—Carbon Footprint Healthcare**	**2022**	**0.4**	**0.84**	**Public and Private**
**United Kingdom—NHS**	2020	31.1	0.47	Public
**Quebec—Association pour la Santé Publique du Québec**	2023	2.7	0.31	Public
**Portugal—Health Care Without Harm**	2014	3.9	0.38	Public and Private
**Australia—Lancet Planetary Health**	2014	35.8	1.52	Public and Private

## Data Availability

The collected datasets presented in this article are not readily available because they precisely represent the activity of each institution. Requests to access the datasets should be directed to each institution separately. The emission factors’ origins are detailed in Table 1. The open-source EFs are included in the Appendix A.

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
