# Peer review of "Estimating the Carbon Footprint of Healthcare in the Canton of Geneva and Reduction Scenarios for 2030 and 2040"

_ijerph, 2024, doi:10.3390/ijerph21060690_

Round 1

Reviewer 1 Report

Comments and Suggestions for Authors

The article addresses a very important topic of carbon footprint of the healthcare sector. The scope of the study is Canton of Geneva, and the analysis is based on data collected from primary sources. However, authors are not able to demonstrate the quality of data used in the study. Few other comments below:

Page 2, Line 72 – Further clarification on the data collection is needed. Please provide details on the number of hospitals, clinics etc. that were contacted. Data used for the analysis should be presented at least in the supplementary information.

Page 3, Fig. 1 – System boundary needs to be clearly defined using a figure and also by stating any major activities or materials that have been excluded from the analysis and justification for doing so.

Page 5, Line 138 – Again, how many hospitals, how many nursing homes? Presented information is not sufficient to derive clear inferences from the results.

Page 6, Figure 4 – Poor chart formatting, y-axis title missing.

Page 7, Figure 5 – Medicines and medical devices is too broad of a category and needs to be separated. Also, it is not clear if the medical devices emission factors are based on a cradle-to-gate analysis or is it just based on material EFs?

Page 7, Line 184 – How is the “ease of application of a measure” quantified and how is the reduction potential calculated?

In the limitations of the study authors should also state that the scope of the study is only limited to GHG emissions, there are other life-cycle environmental impacts from the activities in healthcare that have not been considered.

General editorial feedback - Unclear and wordy sentences throughout the article, please rephrase or break down in shorter sentences.

Comments on the Quality of English Language

Unclear and wordy sentences throughout the article, please rephrase or break down in shorter sentences.

Reviewer 2 Report

Comments and Suggestions for Authors

The manuscript titled ‘Estimating the carbon footprint of healthcare in the canton of Geneva and reduction scenarios for 2030 and 2040’ presents a carbon footprint analysis of the health care sector in Geneva, CH. Overall the manuscript provides useful insights into what is driving emissions in the health care sector in Geneva, but the manuscript needs revising before it can merit publication.

·         The term ‘canton’ is not common in many countries. Can a definition please be given?

·         Fig 1- can it be defined which activities are Scope 1, 2 or 3?

·         Two types of emission factors have been used- unit based (from ecoinvent and other sources) and monetary. The latter is sensitive to inflation as well as general market and global activities. While it is difficult to assess what the discrepancies between the two are, especially when only literature data is being used, could the authors comment on how the two types of emission factors could hinder the results found, especially as monetary EF were used for medicines and medicines were found to be a major source of emissions.

·         Would it be possible to include the EF data used in the SI, if any of the data used were open source?

·         The uncertainties in the results are mentioned in the discussion section of the manuscript but it would be useful to mention this in the results section.

·     In the emission reduction scenarios modelled, do these take into account shifts in the demand for different types of health care? Switzerland, like other countries in Europe, will experience strains and increased demand to its health care sector due to aging population and growing number of people with lifestyle related health ailments (high blood pressure, diabetes etc.).

Comments on the Quality of English Language

The manuscript would benefit from being proofread by a native English speaker as there are some grammatical issues and some issues with tenses throughout the manuscript.

Reviewer 3 Report

Comments and Suggestions for Authors

This study, the first of its kind in Switzerland, deciphers where most of the greenhouse gas emissions arise and proposes action levers to hope to pave the way for ambitious emission reduction policies, if the conclusions are reliable. This paper has relatively detailed survey data, but this study lacks a summary and elaboration of the most core basic methods of carbon emission accounting, a clear and scientific elaboration of the scope of community carbon emission accounting, especially how the relevant activities involved in hospitals should be allocated, etc., and unfortunately we have not seen a good reference to the most core references[8] Global Protocol for Community-Scale Greenhouse Gas Inventories. A scientific and thorough review of carbon emission accounting methods and scopes are suggested, so as to further strengthen the reliability of the conclusions.

Reviewer 4 Report

Comments and Suggestions for Authors

In this manuscript, the authors describe the carbon footprint of the Geneva canton health care system, with future projections and potential measures to reduce GHG emissions. The methodology and approach is consistent and scientifically supported and the whole work is interesting, providing insight into emissions from health care in Switzerland.

However, in the Results section, the authors introduce proposed measures by simply citing them in Table 4 without sufficient detail. The measures are listed in a generic manner without explanation of how they were formulated or the methods used to estimate their impact on reducing emissions. While estimated reductions are provided, the assumptions behind these estimates and the calculation methodology are not explained.

Comments on the Quality of English Language

Generally, English Language is fine. However, a check should be done for minor errors and typos.

Round 2

Reviewer 3 Report

Comments and Suggestions for Authors

1.In the reply of the cover letter, regarding the title, given that the authors collected raw data on the activities of its stakeholders and the analysis shows that when excluding medicines and medical devices hospitals are the main greenhouse gas emitter by far, accounting for 48% of the healthcare system's emission, followed by nursing homes (20%), private practice (18%), medical analysis laboratories (7%), dispensing pharmacies (4%), the homecare institution (3%) and the ambulance services (<1%), it is suggested to add carbon footprint accounting to the title rather than just prediction.

2.The writing logic and review part of the article need to be improved. Particularly, in the method section, the authors should refer to relevant articles[1-2] and add a system boundary diagram for carbon footprint accounting instead of just stakeholder participants, describe relevant accounting methods through standardized carbon emission accounting equations, and classify and elaborate on the specific relevant data collection and carbon emission factor sources for each part, etc. At the same time, the authors has mentioned in the cover letter about “ the same methodology used by the state of Geneva as they share their methodology with the University Hospital, the work emerges from discussion with the University Hospital of Geneva and we have the same data collection survey, most of the same EFs, the same calculation method as them”, however, the pdf file only presents the results, without a specific explanation of why it is classified, and the specific data and calculations process is missing, and these are exactly what need to be reflected in the article.

3.The literature review needs to be expanded to include a broader range of seminal works and current studies on carbon footprint of healthcare.

4.In the scenarios part, this study is about estimating the carbon footprint of healthcare in the canton of Geneva and reduction scenarios for 2030 and 2040, therefore, detailed analysis and scientific explanation of the prediction scenario settings and their specific requirements and illustration are required. It is recommended that the scenario prediction illustration in the “Discussion” part should be firstly elaborated in 2. Materials and Methods of the scenario setting, thereby providing a scientific basis for the final prediction suggestions. In the meantime, please explain in detail the relevant projects that appear in the article, such as Line 162 shift project and NHS.

5.Finally, the manuscript still requires significant improvements in terms of writing clarity, structure, and organization. A thorough proofreading and copy-editing pass is necessary to meet the publication standards.

[1]Rizan C, Bhutta M F, Reed M, et al. The carbon footprint of waste streams in a UK hospital[J]. Journal of Cleaner Production, 2021, 286: 125446.

[2]McAlister S, McGain F, Breth-Petersen M, et al. The carbon footprint of hospital diagnostic imaging in Australia[J]. The Lancet Regional Health–Western Pacific, 2022, 24.
